# A Comparison of Natural and Therapeutic Anti-IgE Antibodies

**DOI:** 10.3390/antib13030058

**Published:** 2024-07-16

**Authors:** Monique Vogel, Paul Engeroff

**Affiliations:** 1Department of Rheumatology and Immunology, University Hosptial of Bern, 3010 Bern, Switzerland; monique.vogel@unibe.ch; 2Department for BioMedical Research, University of Bern, 3012 Bern, Switzerland

**Keywords:** IgE, IgG, Fc receptors, IgE receptors, autoantibodies, IgG-IgE complex

## Abstract

Immunoglobulin E (IgE) plays a critical role for the immune system, fighting against parasites, toxins, and cancer. However, when it reacts to allergens without proper regulation, it can cause allergic reactions, including anaphylaxis, through a process initiated by effector cells such as basophils and mast cells. These cells display IgE on their surface, bound to the high-affinity IgE receptor FcεRI. A cross-linking antigen then triggers degranulation and the release of inflammatory mediators from the cells. Therapeutic monoclonal anti-IgE antibodies such as omalizumab, disrupt this process and are used to manage IgE-related conditions such as severe allergic asthma and chronic spontaneous urticaria. Interestingly, naturally occurring anti-IgE autoantibodies circulate at surprisingly high levels in healthy humans and mice and may thus be instrumental in regulating IgE activity. Although many open questions remain, recent studies have shed new light on their role as IgE regulators and their mechanism of action. Here, we summarize the latest insights on natural anti-IgE autoantibodies, and we compare their functional features to therapeutic monoclonal anti-IgE autoantibodies.

## 1. The Induction of IgE Responses

Immunoglobulin E (IgE) antibodies are thought to be employed by the immune system to fight against a variety of opponents including parasites, toxins, and cancer [1,2,3,4,5]. In contrast, when IgE is produced in response to allergens in the absence of proper control mechanisms, it can trigger pathological type I hypersensitivity reactions including severe anaphylaxis [6,7,8,9]. Several aspects of the IgE response are still poorly understood, such as the mechanisms of IgE mutation, class-switching, and IgE memory formation, which have been summarized and discussed by others [10,11,12]. IgE responses are induced upon antigen presentation in the presence of IL-4- and IL-13-producing T cells, which stimulates the differentiation of B cells into plasma cells that produce IgE. A specific focus has been on T follicular helper and regulatory cells as critical regulators of IgE induction and the inflammatory potency of IgE responses [13,14,15,16]. It has to be noted that IgE induction alone is not sufficient for driving inflammatory potency. IgE is in competition with IgG, which can counteract and suppress IgE-mediated inflammatory effects [17,18]. Thus, allergen-specific IgG may block IgE-dependent effects. In turn, a lack of IgG blocking antibodies may result in increased allergic inflammation [19].

Interestingly, IgG does not only compete with IgE for specific antigen binding, but can also directly bind to IgE, thereby controlling IgE function. These natural IgG anti-IgE autoantibodies have long been observed and investigated but are still not fully understood today. In contrast, basic and translational research has been focused on the generation and application of therapeutic monoclonal anti-IgE antibodies that can suppress IgE-mediated inflammation and thus treat allergic diseases. In this review, we summarize the latest insights on the mechanism-of-action of natural anti-IgE antibodies, and how they compare to monoclonal therapeutic anti-IgE antibodies. We propose that the study of natural anti-IgE antibodies may help us to better understand the functional differences between current therapeutic anti-IgE antibodies and potentially allow for us to even further optimize them. Moreover, this knowledge may translate into the development of polyclonal anti-IgE vaccines as alternatives to current monoclonal antibody approaches.

## 2. Two Major IgE Receptors FcεRI and FcεRII (CD23)

The IgE Fc receptors are critical components in regulating IgE functionality. There are two main types of IgE receptors: the high-affinity IgE receptor (FcεRI) and low-affinity IgE receptor (FcεRII or CD23). FcεRI is primarily found on the surface of mast cells and basophils, and to a lesser extent on other cells such as eosinophils, monocytes, and dendritic cells. Mast cells and basophils, the key initiators of IgE-dependent allergic reactions, express tetrameric FcεRI consisting of one α-, one β-, and two γ-chains. Other cell types express a trimeric form of FcεRI lacking the β-chain, altering FcεRI signaling and function [20]. FcεRI binds to IgE with high affinity. In FcεRI-bound state, IgE can persist for long periods despite its very short serum half-life. Cross-linking of FcεRI by antigen-bound IgE leads to cell activation, which triggers the release of histamine and other inflammatory mediators, responsible for the symptoms of allergic reactions such as asthma, hives, and anaphylaxis [21,22,23]. IgE is the most heavily glycosylated antibody and recent studies have unveiled the functional importance of this glycosylations, in particular, a highly conserved high-mannose glycosylation located at N394 in human IgE and N384 in mouse IgE [24,25,26]. This glycosylation site has been shown to be instrumental in FcεRI binding and activation [27]. Another study demonstrated the importance of IgE sialylation in determining allergic pathogenicity [28].

CD23 (FcεRII) is expressed on a broader range of cells, including B cells, follicular dendritic cells (FDCs),macrophages, and eosinophils. CD23 has a lower affinity for IgE compared to FcεRI and serves multiple functions. Surprisingly, while CD23 is a lectin, its binding of IgE in humans requires calcium but occurs independently of glycans [29,30,31]. Functionally, CD23 can regulate IgE synthesis, clear serum IgE levels, internalize IgE immune complexes (IgE-ICs), and regulate antigen presentation and immune responses [32,33,34]. IgE is not only a strong trigger of immediate inflammation but also a strong trigger of adaptive immune responses. An understudied aspect of IgE biology is its immunogenicity and its function in shaping secondary immune responses, thus acting as a “natural adjuvant” that promotes antigen-specific T cell and antibody responses. IgE-ICs have been shown to induce strong T cell proliferation as well as boost antigen-specific IgG responses [35,36,37].

However, the exact mechanism by which IgE facilitates antigen presentation is still not entirely clear. Evidence suggests that the low-affinity IgE receptor CD23 expressed in B cells has a regulatory role in the process [32]. In certain systems, IgE–antigen complexes are directly internalized by B cells and utilized for the activation of T cells [37,38,39]. Other studies propose that CD23 is more important for the absorption of IgE–antigen complexes and their clearance and transport, whereas antigen presentation occurs independent of CD23 [40,41,42].

Besides membrane IgE Fc receptors, IgE function may be regulated by soluble IgE receptors. Soluble CD23 (sCD23) can be measured in circulation and has been used as a biomarker of disease activity in a variety of conditions, including allergy, cancer, and autoimmunity [43,44]. Soluble FcεRI (sFcεRI) is mostly detected in complex with IgE and elevated serum sFcεRI has recently been used as biomarker in IgE-mediated diseases [45]. Galectin-3 (εBP) is a soluble IgE receptor that binds to IgE and may block FcεRI activation by promoting IgE:FcεRI complex internalization [43,46,47]. Galectin-9 appears to block antigen access to IgE, thereby preventing cell degranulation [47]. Overall, the biological role of the different types of soluble IgE receptors is still a matter of investigation.

## 3. Anti-IgE Therapy with Omalizumab: Blocking FcεRI Function

Anti-IgE therapy is a targeted treatment approach used primarily to manage and mitigate allergic diseases by neutralizing IgE. Omalizumab is currently the only licensed monoclonal antibody that selectively binds to IgE. Omalizumab binds to free IgE, preventing it from interacting with IgE receptors FcεRI and CD23, thereby inhibiting the allergic cascade before it starts. Moreover, it does not cross-link FcεRI-bound IgE but rather accelerates IgE dissociation from FcεRI. These mechanisms help reduce the frequency and severity of IgE-mediated reactions, making omalizumab an effective treatment for conditions such as severe allergic asthma, chronic spontaneous urticaria (CSU), and chronic rhinosinusitis with nasal polyps (CRSwNP) [48,49,50,51,52,53,54].

The development of anti-IgE therapy represents a significant advancement in the treatment of allergic diseases. Continued research is focused on improving the efficacy of these therapies in terms of reducing injection doses and treating non-responders. Interestingly, total IgE levels predict treatment success with omalizumab in CSU [55,56,57]. Another area of research is the application for omalizumab beyond its current indications. A promising recent trial has demonstrated utility for omalizumab in treating food allergies, specifically in preventing allergic reactions to accidentally exposed food allergens, but future studies will need to validate those findings [58,59,60]. Evidence is growing that IgE could be involved in autoimmune diseases such as in bullous pemphigoid (BP) or systemic lupus erythematosus (SLE), which has sparked the evaluation of omalizumab in those conditions [61,62,63]. Another indication for omalizumab could be as an adjuvant compound in specific allergy immunotherapy to improve the safety of allergen injections [64,65,66].

## 4. Omalizumab: Still without Competition?

A variety of alternatives to omalizumab have been developed, but most have so far not advanced in clinical trials. Other excellent reviews have summarized the current and future landscape of novel therapeutic anti-IgE antibodies and other anti-IgE molecules in allergy to which we hereby refer [67,68,69,70]. As interesting potential competitors to omalizumab, we here mainly focus on two emerging monoclonal anti-IgE antibodies: ligelizumab and UB-221 [71,72,73,74]. Ligelizumab, the most clinically advanced alternative to omalizumab, has shown efficacy in phase III studies for CSU [75], whereas for UB-221, phase II studies are currently ongoing for CSU (NCT05298215).

Mechanistically, all anti-IgE monoclonals have the ability of blocking IgE:FcεRI interaction without triggering a cross-link, but interestingly, their effect on the IgE interaction with CD23 is more diverse. As previously mentioned, omalizumab blocks IgE binding to both FcεRI and CD23. In contrast, ligelizumab exhibits a higher degree of overlap with FcεRI, has a higher affinity for IgE, and inhibits the interaction of FcεRI and IgE more efficiently. However, ligelizumab seems to allow for more CD23 interaction than omalizumab, as it can recognize CD23-bound IgE [76]. UB-221 is distinct from omalizumab and ligelizumab, as it fully allows for the interaction of IgE with CD23 and even enables its targeting [72,77]. UB-221 can recognize CD23-bound IgE and preformed complexes of UB-221:IgE can cross-link CD23 [72]. Thus, the more IgE antibodies bind to UB-221, the more they are directed towards CD23, directing IgE to a non-inflammatory pathway by clearing IgE from the serum [32]. Whether ligelizumab or UB-221 can outperform omalizumab in any indication remains to be determined. Ligelizumab has demonstrated safety and efficacy in CSU, but despite initial hope that it could outperform omalizumab, recent clinical trials did not support this hypothesis [73,74,75]. A phase III trial to evaluate the use of ligelizumab in food allergy is currently ongoing (NCT05678959). In summary, further investigations are needed to fully understand the clinical value of these novel alternatives to omalizumab.

## 5. Natural Anti-IgE Autoantibodies: Friends or Foes?

Natural anti-IgE autoantibodies have been described a long time ago [78,79,80,81,82]. However, they are still a mysterious entity today. Over the years, a variety of publications in mice and humans have investigated natural anti-IgE. Given their polyclonal nature, a puzzling question that still occupies researchers is their ability or inability to trigger FcεRI cross-linking and anaphylaxis. Given that these antibodies occur in healthy individuals and thus likely fulfil a physiological role; it is difficult to imagine that they trigger degranulation. In turn, elevated anti-IgE levels are often observed in pathological contexts such as atopic dermatitis or urticaria [83,84,85]. Studies using in vitro degranulation assays have observed that natural anti-IgEs can be anaphylactogenic or suppressive dependent on the setup [85,86,87,88,89]. In contrast, in vivo studies have rather interpreted natural anti-IgE autoantibodies as negative regulators of IgE-dependent inflammation [90,91,92]. Potential explanations for this discrepancy could be the kinetics and concentrations of in vivo-released anti-IgE compared to in vitro stimulation. Other explanations could be the presence of other cell types and/or IgE-regulating co-factors that suppress anaphylaxis in vivo. For IgE itself, the anti-IgE epitope and affinity have also been shown to be important aspects to consider [93,94]. Our own research has pointed towards IgE glycosylation as a key regulator.

## 6. Natural Anti-IgE Autoantibodies: Role of IgE Glycans

We have shown that anti-IgE autoantibodies induced in healthy normal mice immunized with IgE-allergen immune complexes are glycan-specific. Specifically, the same conserved IgE mannose region that is essential for FcεRI binding is recognized by natural anti-IgE [92]. We then evaluated how this immunogenicity of IgE-allergen IC shapes secondary responses in an allergic model. While anti-allergen IgG responses were elevated in these mice, anti-IgE antibodies were likewise increased. This led to a reduction in IgE levels and significantly reduced allergy and systemic anaphylaxis in mice [92]. The removal of glycans from IgE significantly disrupted its ability to promote neutralizing IgG anti-IgE autoantibody response, thus reducing serum and basophil IgE levels and suppressing allergy [95]. Of note, human IgG anti-IgE autoantibodies, are likewise mostly glycan-specific [92]. Our findings suggest that anti-IgE autoantibodies modify the Fc receptor pathway as IgG-IgE complexes are increasingly bound and absorbed by low-affinity receptors CD23 and FcγRs while binding to FcεRI is suppressed (Figure 1).

Most recently, we translated these findings into a more clinically feasible approach and developed an anti-IgE vaccine based on virus-like particles (VLP) displaying IgE-Cε fragments [96]. Similar to IgE–allergen ICs, this vaccine reduced IgE levels and allergic symptoms in mice without causing any side effects [96]. Overall, our research suggests that FcεRI and natural anti-IgEs compete for the same conserved single IgE mannose glycosylation site [77,97,98,99]. Hence, glycan specificity could be a key attribute for suppressive function of natural anti-IgEs. The competition between FcεRI and natural anti-IgE autoantibodies for the same conserved mannose on IgE could likewise be a good explanation for their non-anaphylactogenic nature (Figure 2). Nevertheless, other mechanisms might be at play, especially for non-glycan-specific anti-IgE autoantibodies.

## 7. A Mechanistic Comparison of Natural and Therapeutic Anti-IgE Antibodies

Natural anti-IgE autoantibodies are overall still understudied but recent results from humans and mice demonstrate a striking uniformity of glycosylations as essential regulators of IgE function [24,25]. Still, it is important to note that these studies primarily involve a polyclonal response, and non-glycan-specific anti-IgE autoantibodies may also be present. Although it is difficult to compare a polyclonal response to monoclonal antibodies, we showed that IgE recognition by omalizumab is in fact dependent on the same N394 mannose structure on IgE as the majority of natural anti-IgE antibodies [100]. In contrast, omalizumab blocks CD23 interaction, which is not the case for most natural anti-IgEs. In fact, polyclonal natural anti-IgE antibodies favor CD23 binding, which is an important contributor to the absorbance/and serum clearance of IgG-IgE complexes. Ligelizumab is closer to natural anti-IgE antibodies than omalizumab in terms of its interaction with CD23, while its glycan dependency is not yet known. Nevertheless, it does not seem to enhance the targeting of CD23. In contrast, the final anti-IgE, UB-221, is very similar to natural anti-IgEs in terms of CD23 binding and may even be enhanced by its presence. However, we do not yet know the glycosylation dependency of this antibody (Table 1).

## 8. Lessons from Anti-IgE on the Role of CD23 in Immunopathology

The different mechanistic and functional interaction of those three monoclonal anti-IgE antibodies and natural anti-IgE antibodies with CD23 could allow for us to speculate on the role of CD23, which is still not fully understood. However, the simultaneous binding of Fcγ receptors should not be ignored and could also explain the clinical differences in those antibodies [101]. The recent findings on ligelizumab and UB-221 suggest that allowing for CD23 interaction may not be problematic in CSU. It has even been argued that CD23-mediated IgE clearance could be a beneficial feature of anti-IgE therapy [72]. In mice, CD23 is important for the non-inflammatory clearance of IgG-IgE complexes via natural anti-IgE autoantibodies [92,95]. The specific advantage of omalizumab in allergic asthma raises the question about the importance of CD23 blockade in this condition. Studies using CD23-blocking antibodies have proposed that CD23 is involved in allergic airway inflammation [102,103]. 

In contrast, CD23-deficient mice develop increased allergic airway inflammation, demonstrating that CD23 expression is not required [102,103,104,105]. A potential reason for these contradicting results could be the effect of CD23-blocking antibodies and the experimental setup. Future studies will need to consider the cell-specific expression of CD23, as it can be expressed in a variety of cells including B cells, FDCs, epithelial cells, and even airway smooth muscle cells [102,106,107,108,109,110]. Moreover, two CD23 isoforms exist, for example monocyte-related cells express CD23b but not CD23a while B cells constitutively express more CD23a [111,112,113]. Although the difference between the two CD23 isoforms is only a few amino acids in the intracellular domain, CD23a targets IgE complexes to an endocytosis and recycling pathway whereas CD23b targets IgE complexes to a phagocytic degradative pathway [41,112,113,114,115,116].

Overall, whether CD23 blockade is an important feature for omalizumab in the treatment of allergic asthma remains to be determined. It needs to be emphasized that the effect of IgG-IgE complexes and their interaction with CD23 and Fcγ receptors have largely not been investigated in the context of allergic asthma.

## 9. The Functional Effects of IgG-IgE Complexes: The Interesting Case of IgE Clone SPE-7

We have begun to understand that natural anti-IgE autoantibodies downregulate IgE levels because IgG-IgE complex formation accelerates the engagement of low-affinity Fc receptor and hence serum clearance. Nevertheless, the biological effects of IgG-IgE complexes compared with free IgE or free IgG on various Fc receptor-expressing immune cells remain to be studied in more detail. However, interestingly, some of these effects may have been studied inadvertently. We recently noticed that one of the most studied mouse IgE monoclonals, the hybridoma-produced IgE clone SPE-7 contains mouse IgG that forms complexes with IgE [117]. Over the years, SPE-7 has shown some interesting functional characteristics compared to other IgE clones. SPE-7 was thought to have “cytokinergic” function, the ability to activate signaling in mast cells in absence of a cross-linking antigen, resulting in a variety of effects including histamine/leukotriene release, cytokine release, and an increase in mast cell survival [118,119,120,121]. In other studies, SPE-7 was shown to engage IgG receptors [122,123]. We have now shown that many of these distinct features for SPE-7 are removed by purifying the IgG-IgE complexes from the preparation. Similarly, we observed increased CD23 binding by non-purified SPE-7, which is in line with previous findings on natural anti-IgE autoantibodies. An interesting difference between IgE SPE-7 and classical IgE–antigen activation in mast cells was reported to be the signaling kinetics, which were slower for SPE-7. Additionally, histamine and leukotriene release was lower. Interestingly, the survival of mast cells was increased, a feature not observed with classical IgE–antigen activation (Figure 3) [120,121]. The extent to which all these mechanisms occur in vivo remains to be determined.

## 10. Therapeutic IgE Antibodies against Cancer: Considering Natural Anti-IgE

Therapeutic monoclonal antibodies of the IgE isotype have emerged as a promising area of research in cancer treatment. Unlike conventional therapies that focus on suppressing or modulating immune responses, monoclonal IgE antibodies in cancer therapy leverage the potent immune-activating properties of IgE to target and destroy cancer cells. It is thought that the binding of IgE to its receptors can lead to rapid and robust cell-mediated cytotoxicity against tumors. While the use of therapeutic IgE antibodies in cancer is still largely experimental, several preclinical studies have shown promising results. Moreover, studies have demonstrated that IgE therapeutics do not induce type I hypersensitivity reactions, thereby providing evidence for the safety of IgE antibody immunotherapy for cancer [124,125,126,127,128].

As mentioned previously, natural anti-IgE antibodies are present in individuals without any external intervention. In the context of cancer therapy, therapeutic IgE antibodies might interact with these natural anti-IgE antibodies, which may influence the effectiveness and outcomes of therapy. Monitoring patients receiving IgE-based cancer therapies for the presence of anti-IgE antibodies might help to predict and manage impacts on treatment efficacy. Further research is needed to determine the role of natural anti-IgE antibodies in monoclonal IgE anti-cancer therapy.

## 11. Conclusions

In conclusion, anti-IgE therapies have shown promise in treating IgE-driven diseases by clearing serum IgE and disrupting IgE interactions with FcεRI without triggering the allergic reaction. Interestingly, therapeutic anti-IgE antibodies have variable effects on CD23 interaction. Recent studies have highlighted the physiological role of natural anti-IgE antibodies, which neutralize IgE and reduce serum and FcεRI-bound IgE levels. These natural anti-IgE antibodies recognize conserved mannose structures on IgE and facilitate CD23 and FcγR interaction, which contributes to the rapid serum clearance of IgG-IgE complexes. The future study of natural anti-IgE antibodies and IgG-IgE complexes may offer valuable insights into physiological IgE regulation which could enable an optimization of anti-IgE therapies for improved efficacy and clinical impact in a variety of IgE-mediated immunopathological conditions.

## Figures and Tables

**Figure 1 antibodies-13-00058-f001:**
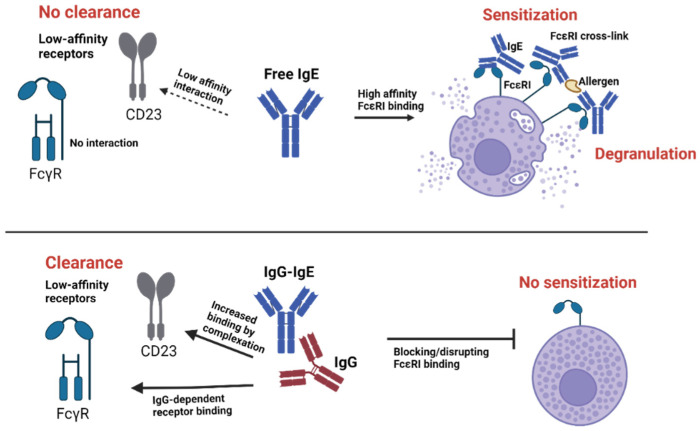
**Natural IgG anti-IgE in the regulation of FcεRI.** Free IgE binds to FcεRI with high affinity. The binding of IgE by natural anti-IgEs suppresses FcεRI targeting by favoring the targeting of low-affinity IgE receptors due to increasing avidity interactions. Moreover, the binding to FcγRs is unlocked by the presence of IgG. Created with BioRender.

**Figure 2 antibodies-13-00058-f002:**
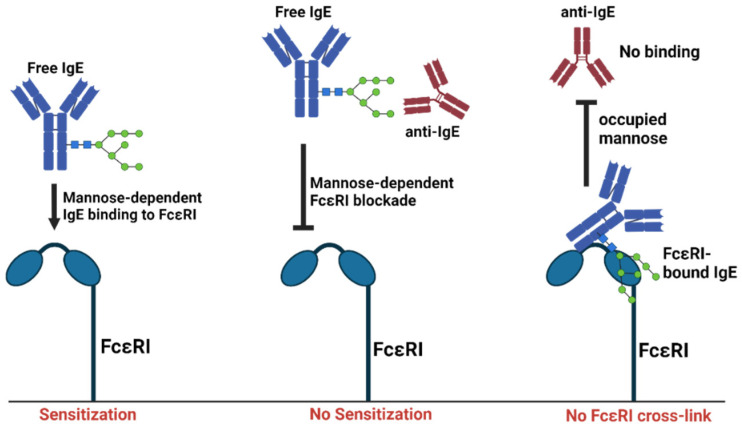
**IgE mannose competition as a regulator of FcεRI sensitization.** Free IgE binds to FcεRI in mannose-dependent fashion (N394 in human IgE and N384 in mouse IgE) leading to sensitization. The majority of natural anti-IgE autoantibodies recognize the same mannose region. Thus, IgE binding to FcεRI is blocked by natural anti-mannose IgE antibodies. For FcεRI-bound IgE, access to mannose epitopes is blocked, which may prevent the anaphylactic responses to natural anti-IgE autoantibodies. Created with BioRender.

**Figure 3 antibodies-13-00058-f003:**
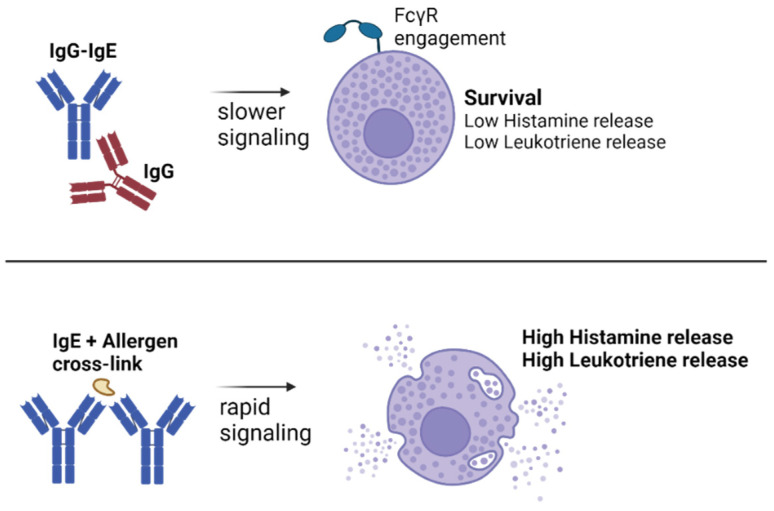
**Lessons from IgE clone SPE-7 on the mechanisms of IgG-IgE complexes.** The antigen-independent activation of mast cells by “cytokinergic” IgE clone SPE-7 is explained by the presence of IgG-IgE complexes [117]. Compared to classical IgE + allergen degranulation, “cytokinergic” IgE was shown to trigger slower signaling, a lower release of granules, but increased mast cell survival [121]. Created with BioRender.

**Table 1 antibodies-13-00058-t001:** A comparison of natural versus therapeutic anti-IgE antibodies.

	Omalizumab	Ligelizumab	UB-221	Natural Anti-IgE
Clonality	Monoclonal	Monoclonal	Monoclonal	Polyclonal
In vivo FcεRI interaction	Inhibition	Inhibition	Inhibition	Inhibition
In vitro FcεRI interaction	Inhibition	Inhibition	Inhibition	Inhibition/activation
CD23 interaction	Inhibition	Partial Inhibition	Promotes binding	Promote binding
IgE-glycan-dependent	Yes	?	?	Mostly yes

## Data Availability

No new data were created or analyzed in this study. Data sharing is not applicable to this article.

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
