# Peer review of "A Comparison of Natural and Therapeutic Anti-IgE Antibodies"

_2073-4468, 2024, doi:10.3390/antib13030058_

Round 1
Reviewer 1 Report
Comments and Suggestions for Authors
Topic : synthesizing the up-to-date knowledge on natural and therapeutic anti-IgE antibodies.
This is a great review that will surely be worth of interest among specialist in the field. Though, I would suggest minor recommandations :
1. INTRODUCTION
- The following sentence line 37-38 « Thus, low IgG responses combined with elevated IgE re- 37 sponses are required for IgE-biased responses that can result in allergic inflammation. » is unclear for me. Could the author reformulate ?
2. Two major IgE receptors FcεRI and FcεRII (CD23)
- Lines 54-56 : « FcεRI is primarily found on the surface of mast cells and 54 basophils, and to a lesser extent on other cells such as eosinophils, monocytes, and dendritic cells. FcεRI binds to IgE with high affinity, and its cross-linking by antigen-bound IgE leads to cell activation ». Could the author mention that that FceRI on others cells than mast-cells and basophils is a trimer instead of tetramer. Thus, it engages different signaling and function.
- Line 65 :
- Could the authors precise in which pathology sCD23 could be a biomarker ?
- Concerning gelectins, the authors should briefly quote which galectins are interacting with IgE (ex : Galectin 3 etc…) and how it modulates its function ?
- Lines 74-75 : « Surprisingly for CD23, despite it being a lectin, its binding of IgE in humans seems to occur independent of glycans ». I would precise how CD23 and IgE are interacting since glycan are not in the game indeed.
3. Anti-IgE therapy: Blocking the sensitization of FcεRI
- Line 109 : Careful typo « Blockins »
- Line 121 : » Omalizumab seems to be best in asthma… ». This sentence is a bit ambiguous even though I see what the authors meant. Indeed, Omalizumab is highly effective in allergic asthma and not to other asthma phenotypes (eosinophilic etc…).
4. Natural anti-IgE autoantibodies: Friends or foes?
- Lines 130-136 : « Given that these antibodies occur in healthy individuals and thus likely fulfil a physiological role; it is difficult to imagine that they trigger degranulation. In turn, elevated anti-IgE levels are often observed in pathological contexts such as atopic dermatitis or urticaria. Studies using in vitro degranulation assays have observed that natural anti-IgEs can be anaphylactogenic or suppressive dependent on the setup. In contrast, in vivo studies have rather interpreted natural anti-IgE autoantibodies as negative regulators of IgE-dependent inflammation ». This is a very interesting part. Could the authors speculate/hypothezise why such discrepencies are observed between natural anti-IgE ? Could it be glycan ? Could it be affinity to IgE receptors ?
6. A mechanistic comparison of natural and therapeutic anti-IgE antibodies
- Line 160 : « Nevertheless, recent results from humans and mice demonstrate a striking uniformity of glycan-specificity ». This sentence needs proper quotations.
- Table needs a legend and a table number.
7. Lessons from anti-IgE on the role of CD23 in immunopathology
- Lines 188 - 191 : « Moreover, two CD23 isoforms exist, as monocyte-related cells express CD23b but not CD23a whereas B cells constitutively express more CD23a. Although the difference between the two CD23 isoforms is only a few amino 190 acids in the intracellular domain, CD23 targeting in different cell types such as macrophages may trigger inflammatory responses ». Could the authors briefly cite what are the known functions of each isoforms of CD23 ?
8. The functional effects of IgE-IgG complexes: The interesting case of IgE clone SPE-7
This section would worth a sketch and a small conclusion on the suspected functional effects of IgG-IgE complexes
9. Therapeutic IgE antibodies against cancer: Considering natural anti-IgE
- Lines 219-221 : « Moreover, studies have demonstrated that IgE therapeutics do not induce type I hypersensitivity reactions, thereby providing evidence for the safety of IgE antibody immunotherapy for cancer » . Could the authors state hypothesis of non anaphylactogenic nature of natural anti-IgE ?
Author Response
We thank the reviewer very much for carefully reading our manuscript. We completely agree with all the points raised by the reviewer and have reformulated, corrected, and expanded on each point in the manuscript, according to the reviewer's suggestions. We have also added a figure as the reviewer suggested.
Regarding Point 9, we were unsure if we understood the question correctly, but we rewrote the section.
Reviewer 2 Report
Comments and Suggestions for Authors
Review of the manuscript antibodies-3004351
This is a very interesting account of the features and properties of natural and therapeutic anti-IgE antibodies. The authors provided a deep insight into these topics using appropriate and mostly recent bibliography. Review of natural anti-IgE is a valuable content in this manuscript, through which healthcare professionals more familiar with using anti-IgE’s as a treatment modality for allergies and CSU can acquaint themselves with other aspects of that isotype of immunoglobulins.
May main comment regards the information on therapeutic anti-IgE efficacy in allergic disease and CSU. In line 122 (page 3) the authors state that ligelizumab may outperform omalizumab with regard to CSU treatment. In my belief, this remains in contradiction with data form clinical trials (e.g. PEARL-1 and -2, just to cite most recent ones) showing that ligelizumab and omalizumab were equally effective in CSU and no superiority of ligelizumab vs omalizumab was shown in subjects with CSU.
In general, I suggest expanding a little bit the paragraph discussing the efficacy of anti-IgE in allergy and CSU, including comments on the results of trials investigating efficacy of omalizumab, ligelizumab (and possibly other molecules). Additionally, in my opinion, the statement contained in line 123 about future need of tailor-made anti-IgE therapy based on specific IgE-mediated diseases looks too far-fetched and premature, as far as currently available clinical data are concerned. I would suggest rephrasing this or providing data to support such view.
Otherwise, I have no concerns to raise with regard to this manuscript.
Author Response
We thank the reviewer very much for taking valuable time to read the manuscript carefully. The reviewer is right that we missed some important results of more recent clinical trials for the two antibodies. We have extensively worked on these aspects, addressed these novel insights throughout the manuscript, and expanded the clinical part of the review according to the reviewer's suggestions.
Reviewer 3 Report
Comments and Suggestions for Authors
In the present review, Monique et al., have elucidated a summarized comparison of the functional characteristics of natural anti-IgE autoantibodies (present naturally in healthy individual) and commercially available therapeutic monoclonal anti-IgE antibodies. It is well described in many previous studies that both types of anti-IgE antibodies bind to IgE and inhibits its interaction with the high and low affinity IgE receptor (FceRI) expressed on the surface of mast cells and basophils. This interaction is crucial because the binding of IgE to FceRI is a key step in the allergic response, leading to the degranulation of these immune cells and the subsequent release of histamines and other inflammatory mediators. By blocking this binding, anti-IgE antibodies can effectively prevent the degranulation process and mitigate allergic reactions. This therapeutic approach has been successfully utilized for many years in the treatment of various allergy related disease, providing relief to the patient suffering from conditions such as asthma, and other IgE-mediated disorders. Currently, several commercial therapeutic monoclonal anti-IgE antibodies are available (Omalizumab, Ligelizumab, UB-221), each designed with unique functional properties to target different aspects of the allergic conditions. For example, Omalizumab is designed to block IgE binding to both low and high receptor while UB-221 inhibits the interaction of IgE with high receptor but not with low receptors. So, it is important to know that these therapeutic antibodies differ in their mechanisms of action. Therefore, careful consideration is necessary when selecting an appropriate treatment option to ensure maximum benefit to the patient without any side effects. The present review describes the mechanism by which natural anti-IgE autoantibody’s function and compare them with the mechanisms of monoclonal therapeutic anti-IgE antibodies. This comparison highlights the potential advantage of each approach. Overall, this review article is well-written and understandable. To enhance the manuscript further before publication, I have a few suggestions:
1. Discuss in short on the clinical applications-provide few examples of clinical scenarios where natural and therapeutic anti-IgE antibodies are used, including any recent emerging therapies.
2. Include few more diagrams-Elaborate the molecular mechanisms of action for both types of antibodies with illustrative diagrams.
3. Comparative efficacy and safety-Include a section comparing the efficacy and safety profiles of these antibodies, supported by clinical trial data.
4. Future directions-Discuss potential innovations in the field of anti-IgE antibody therapy, including any ongoing clinical trail or new development in this field.
5. Elaborate the section “Therapeutic IgE antibodies against cancer: Considering natural anti-IgE” Page -5
6. Page 5 line 207…We now showed that these distinct feature….-What authors want to say here is not clear.
7. Page 1-line 39-42-Please add references here.
8. Page 2-line 91-94-Please add references here.
9. Authors have used the term CSU in few places in the manuscript (Page 3, line 122. Page 5, line-181 and line -184). What is CSU. Please elaborate it.
10. Page 3, line 109, Sentence looks grammatically incorrect-It should be ‘Omalizumab binds to the free IgE, preventing it from interacting with both high-affinity IgE receptor (FceR1) and the low-affinity IgE receptor CD23 (FceRII)’.
By incorporating these suggestions, the manuscript can provide an even more thorough and valuable resource in the field of allergy therapy.
Comments on the Quality of English Language
Quality of English is fine.Minor editing is required.
Author Response
We thank the reviewer very much for taking valuable time to carefully read our manuscript. We appreciate the kind feedback and agree with every comment made by the reviewer. We have incorporated changes to all points in the manuscript.
Round 2
Reviewer 1 Report
Comments and Suggestions for Authors
Thanks to the authors for replying to the comments.
Thanks to the authors for this very interesting work.
In my opinion, the manuscript can be published in its present form.
Reviewer 2 Report
Comments and Suggestions for Authors
The authors answered to my comments and concerns and expanded the manuscript text with regard to clinical aspects that had been raised in the review report. This manuscript is - in my opinion - very interesting and may be appealing and attractive to allergists wishing to broaden their knowledge on the aspects of IgE-driven response that are less familiar and obvious to a clinician.
Reviewer 3 Report
Comments and Suggestions for Authors
I have checked the revised manuscript and find that the authors have addressed all my comments well. The changes and improvements made in this version significantly enhance the significance of the work. I am satisfied with the current improved version of the manuscript and am pleased to recommend this manuscript for acceptance.